# IgY Antibodies as Biotherapeutics in Biomedicine

**DOI:** 10.3390/antib11040062

**Published:** 2022-09-29

**Authors:** Diana León-Núñez, María Fernanda Vizcaíno-López, Magdalena Escorcia, Dolores Correa, Elizabeth Pérez-Hernández, Fernando Gómez-Chávez

**Affiliations:** 1Laboratorio de Enfermedades Osteoarticulares e Inmunológicas, Sección de Estudios de Posgrado e Investigación, ENMyH—IPN, Av. Guillermo Massieu Helguera 239, La Escalera, Gustavo A. Madero, Ciudad de México 07320, Mexico; 2Departamento de Medicina y Zootecnia de Aves, Facultad de Medicina Veterinaria y Zootecnia-UNAM, Ciudad de México 04510, Mexico; 3Dirección de Investigación/Centro de Investigación en Ciencias de la Salud, Facultad de Ciencias de la Salud, Universidad Anáhuac México, Naucalpan de Juárez 52786, Mexico; 4Maestría y Doctorado en Ciencia y Tecnología de Vacunas y Bioterapéuticos, Instituto Politécnico Nacional, Ciudad de México 07738, Mexico

**Keywords:** IgY, biotherapeutics, diagnosis, treatment

## Abstract

Since the discovery of antibodies by Emil Von Behring and Shibasaburo Kitasato during the 19th century, their potential for use as biotechnological reagents has been exploited in different fields, such as basic and applied research, diagnosis, and the treatment of multiple diseases. Antibodies are relatively easy to obtain from any species with an adaptive immune system, but birds are animals characterized by relatively easy care and maintenance. In addition, the antibodies they produce can be purified from the egg yolk, allowing a system for obtaining them without performing invasive practices, which favors the three “rs” of animal care in experimentation, i.e., replacing, reducing, and refining. In this work, we carry out a brief descriptive review of the most outstanding characteristics of so-called “IgY technology” and the use of IgY antibodies from birds for basic experimentation, diagnosis, and treatment of human beings and animals.

## 1. Introduction

The origins of “IgY technology” date back to 1893, when Klemperer et al. carried out an experiment in which they demonstrated that the immunoglobulin from egg yolk is capable of conferring protection against bacterial toxins [1]. For almost 100 years, there was no significant progress in this area. However, since the 1980s, the use of IgY technology has increased due to the development of therapies and diagnostic tests based on these specific antibodies, conjugated with classical markers, such as fluorescein, alkaline phosphatase, and peroxidase [2]. Currently, more than a thousand scientific articles related to IgY technology can be found, with exponential growth since 2008. Researchers from China, the United States, Canada, Japan, and Germany have generated the largest number of publications [3]. In addition, this technology could be implemented with relative ease in developing countries, such as Mexico, due to the high levels of bird production and extensive experience in their management, not only in the veterinary field but also in human medicine for the diagnosis and treatment of infectious diseases and cancer [4,5,6].

## 2. Immunity in Birds

The immune system of birds, like that of humans, is classified into innate and adaptive, both with humoral and cellular components that interact with each other. The immune system consists of barriers, such as the skin and mucous membranes, and primary and secondary lymphoid organs. The bursa of Fabricius and the thymus are primary immune organs in which B and T lymphocytes develop, respectively. Secondary organs include the spleen, Harder’s gland, lymph nodes, bone marrow, and mucous-associated lymphoid tissues (MALTs), such as gut-associated (GALTs) and bronchus-associated lymphoid tissues (BALTs) (Figure 1). Cells involved in the innate response in birds include heterophils (cells analogous to mammalian neutrophils), macrophages, mast cells, eosinophils, and natural killer (NK) cells. They also have dendritic cells, which, similar to macrophages and B lymphocytes, function as antigen-presenting cells. Regarding humoral components, there are the complement system, antimicrobial peptides, and various cytokines, in addition to the antibodies produced by B lymphocytes, which are the subject of interest in this review [7]. Bird B cells produce three immunoglobulins analogous to those of mammals: IgM and IgA, which are transferred to the egg white, and IgY, which is specifically delivered to the yolk [8]. The transfer of IgY from the blood of birds to the egg yolk is mediated by a selective transport mechanism to maturing oocytes and ovarian follicles [9,10].

In egg yolk, IgY reaches levels similar to those in serum (6–13 mg/mL). IgM and IgA arise in the egg white when the egg passes through the oviduct and is transferred to the embryonic canal, while IgY circulates from the blood to the yolk. During embryonic development, the chick absorbs part of the immunoglobulin Y, which passes into its circulation, while the maternal IgM and IgA diffuse into the amniotic fluid and are degraded, so they must be produced by the new organism later in life [7,8,9].

## 3. Structural Characteristics of IgY

IgY is present in birds, reptiles, amphibians, and lungfish. It is considered the evolutionary precursor of IgG and IgE and is functionally equivalent to IgG in mammals; in fact, it has been proposed as the “avian IgG” [2,11]. However, these two Igs present important structural and functional differences, starting with their molecular weight, since IgY weighs 180 kDa and IgG around 150 kDa (IgG3 is slightly larger). Both are composed of two light chains and two heavy chains, but IgY has four Immunoglobulin (Ig) domains in the constant region of its heavy chain, while IgG has only three besides the hinge region, showing that IgY has the same domain architecture as mammalian IgM and IgE. IgY lacks the hinge region, which limits its flexibility (Figure 2), making it highly specific and more resistant to proteolytic degradation [2,7,12]. Several reports support that IgY increases the phagocytosis of specific pathogens by mammalian cells through a mechanism independent of Fc receptors, because IgY does not bind to them [13]. On the other hand, IgY retains its structural stability at a pH of 3.5–11.0 and a temperature of 30–70 °C. It can be preserved for up to one month at room temperature and for years at 4 °C, which could represent an advantage for its use as a biotechnological reagent in diagnosis and treatment [14,15,16].

## 4. IgY in the Diagnosis and Treatment of Diseases

In recent years, IgY has attracted considerable attention as it has advantages over mammalian IgG. For example, IgY can be isolated in larger quantities from the egg yolks of immunized hens as compared to IgG obtained from mammalian blood, which reduces, refines, and may allow replacement of the use of farm animals in experimentation [16,17,18,19]. 

The structural features and production process of IgY have shown some advantages over mammalian IgG, including: (1) the strongest immune response against epitopes of conserved mammalian proteins, which are generally not immunogenic in other members of this family due to their high homology; (2) less cross-reactivity; (3) avoidance of human complement activation; (4) reduction of false positives due to the lack of rheumatoid-factor binding sites in the Fc region; (5) toxic side-effects not having been reported for therapeutic uses; and (6) low-cost extraction, favoring its use as an immunoreagent in diagnosis and treatment [14,15,16,17,18,19,20,21,22,23,24].

IgY antibodies obtained from poultry and purified from egg yolk have been tested as tools for diagnosis and treatment of cancer, as well as infectious, autoimmune, and allergic diseases [5,6,18,25,26,27]. In particular, they have been used against viruses and bacteria which cause respiratory disorders in humans and animals [12]. Due to this, during the recent pandemic caused by the SARS-CoV-2 virus, several efforts were reported worldwide to implement specific monoclonal and polyclonal IgY antibodies for diagnosis and transfer of passive immunity to treat and prevent the infection in a fast and safe way [5,28,29,30,31]. Other examples of the use of IgY against viruses are those related to the management of Middle East respiratory syndrome (MERS) [32], rotavirus [33], influenza [34], Zika [35], dengue [36], and Ebola [37]. IgY antibodies against the MERs S1 protein were able to neutralize virus entry into green monkey (Vero) cells in culture and decrease inflammation in a mouse model of the disease [32]. In the case of rotavirus, there have been many more advances, since specific IgY has been tested in animals and humans and has been shown to reduce diarrhea caused by the virus and even by bacterial pathogens that co-infect neonates. In fact, there are infant milk formulations that include these antibodies which have been on sale since 2015 in Japan and the United States [33]. In the case of the influenza virus, IgY antibodies against the avian H5N1 and H9N2 viruses are effective for neutralization. Therefore, its therapeutic use in birds and humans has been suggested [34]. The lack of a mammalian Fc-receptor-binding region in IgY has been shown to be a therapeutic alternative to the immunological phenomenon known as antibody-dependent enhancement (ADE). ADE causes an increase in viral load because the host can produce low-affinity non-neutralizing anti-virus antibodies which act as “Trojan horse”, allowing the pathogen to enter phagocytic cells, providing the virus with an optimal niche for its development. The use of IgY from chickens or geese against Zika and dengue viruses without inducing ADE has been reported in vitro and in vivo in mouse and goose models [35,36]. Another advantage that may be decisive for the extensive implementation of IgY-based technology treatments is that they are resistant to heat, maintaining their neutralizing capacity for up to a year at 25 °C, which facilitates their distribution to regions of the world such as Africa for the treatment of viruses, including Ebola [37].

Bacteria have the capacity for autonomous replication and can secrete virulence factors, such as toxins, genetic material, and extracellular vesicles, which contain any of the abovementioned components, allowing them to colonize and proliferate in different human and animal cells and tissues [38]. In this context, IgY has been used as a therapeutic option since it can reduce and prevent bacterial colonization. For example, specific IgY against *Escherichia coli* enterotoxin has been proposed as a food-fortification element, since it resists heat-pasteurization at 65 °C for 15 min without affecting its neutralization activity, maintaining the ability to inhibit in vitro the growth of *E. coli* [39,40,41]. 

An interesting study was carried out on *Helicobacter pylori* infection, a pro-carcinogenic bacterium that infects the human stomach and releases vacuolating cytotoxin A (VacA), which favors its persistence. Oral administration of IgY against this toxin has been reported to have a protective effect against *H. pylori* colonization and limits the histological changes induced in the gastric tissue [42]. 

Specific IgY has also been used against pathogenic bacteria of the oral cavity, including *Streptococcus mutans* and *Porphyromonas gingivalis*; published works have shown that these antibodies can inhibit the development of caries, accumulation of dental plaque, and reinfections [17,43,44]. Furthermore, IgY has also helped in the treatment of other oral pathologies, such as periodontitis and halitosis [45]. 

IgY has also been proposed for the diagnosis of infections caused by *Staphylococcus aureus* and to limit skin alterations caused by staphylococcal protein A (SpA), as IgY against SpA is being added to creams and cosmetics to prevent infection [46]. In addition, IgY has been proposed against bacteria such as *Staphylococcus epidermidis*, related to surgical nosocomial infections [38,47], and even against tuberculosis, managing to counteract the bacterial load of the respiratory tract and lungs [48]. 

The use of monoclonal IgY antibodies conjugated with phthalocyanine—a synthetic photosensitizing dye used in near-infrared phototherapy—was recently described for use against the pathogenic opportunistic yeast *Candida albicans*. This IgY immunophototherapy proved to be highly effective and specific in an in vivo skin infection model and caused no damage to the healthy epithelium. This novel mode of specific elimination of microorganisms could be used for anti-infection treatment, changing only the specific antibody and probably limiting the adverse effects of the indiscriminate use of antibiotics owing to selection for resistant variants [49]. Similarly, the therapeutic potential of IgY has been explored in diseases caused by fungi, such as dermatophytosis, due to the fungus *Trichophyton rubrum* [50], and keratitis, caused by *Aspergillus fumigatus*—infections that, according to the World Health Organization, are serious global health problems with high morbidity rates worldwide [50].

In the field of parasitology, IgY has been implemented mainly in the diagnosis of various diseases, such as clonorchiasis, schistosomiasis, and toxoplasmosis; this immunoglobulin has been used in immunoassays to detect circulating parasitic antigens and host antibodies [18]. 

IgY has also been proposed as an antitumor agent due to its ability to induce apoptosis in cancer cells when this immunoglobulin is directed against TRAIL-2R [51]. Furthermore, it has been suggested as a novel immunotherapeutic molecule against carcinogenic stem cells in the treatment of glioblastoma with IgY anti-CD133—a characteristic marker of these cells [6]. 

Finally, IgY has been used to reduce the availability of allergens in companion animals, such as cats. To limit allergic responses, foods are supplemented with IgY antibodies directed against Fel d1—a glycoprotein produced by the salivary and sebaceous glands of cats, considered the primary allergen [26]. 

This technology has also been proposed for the treatment of patients with autoimmune diseases, such as celiac disease, by oral administration of IgY antibodies against gliadin—a gluten protein responsible for the allergic response [27].

## 5. IgY Production and Purification

IgY antibodies can be produced in poultry birds, predominantly in hens, but also in turkeys, quails, ducks, geese, and ostriches [52]. The production of IgY is low-cost and allows high yields when obtained from egg yolk [31]. It has been reported that the median content of IgY per egg is around 60 mg, and some calculations suggest that 50 immunized hens could produce up to 1 kg of antibodies in one year [53]. The quantity of the purified form is directly proportional to the size of the egg, ostrich eggs being the largest, weighing around 1.5 Kg, and quail eggs the smallest, weighing about 10 g. Other factors must also be considered, such as housing, egg-generation time, and egg yield per year [54]. Due to these requirements, chickens are still preferred for IgY production; more recently, the generalized use of chickens has prompted the design of a transgenic strain for IgY antibodies containing the human variable regions and the bird constant regions. This strategy allows the obtention of highly specific IgY antibodies against mammalian conserved proteins, the human variable regions of which can be subsequently cloned into human constant-region genes to produce monoclonal antibodies for safe use in therapy [55].

IgY antibodies can be purified in the laboratory by different strategies, such as water dilution, PEG precipitation, and treatments with anionic polysaccharides, organic solvents, and other specific chemicals, though all of these strategies face the challenge of lipid removal from the yolk. For antibodies intended to treat human diseases, the purities required are high and the characterizations significant, especially the structure, charge state, and microheterogeneity, each of which must be determined by at least two analytical methods [56,57]. Since *S. aureous* A or G proteins, which bind mammalian antibodies, are ineffective for IgY purification, different ligands have been proposed for the affinity chromatography purification of pharmaceutical use [57,58]. Protein M—a transmembrane protein from human mycoplasma—has been reported as the most effective IgY binding protein, allowing 98.7% purity of the whole molecule and 88.7% and 90.1% for Fab and Fc post-papain digestion fragments, respectively [58,59]. Despite these promising data, no evaluation of IgY impurities, such as endotoxin, have been reported.

## 6. Pharmacokinetics and Pharmacodynamics of IgY

Information related to the pharmacokinetics of IgY is currently scarce. The available data come from animal models with the application of sublethal doses of xenogeneic snake venom [60]. In these studies, IgY was shown to have a wide and rapid distribution in the animal body, including muscles and brain parenchyma [61]. The main route of elimination was hepatobiliary (fecal). Accumulation of antivenom IgY in the muscle fascicles and abdominal regions of animals injected with venom suggested the formation of specific antigen–antibody complexes shortly after intravenous administration which completely blocked venom-induced paralysis within 3 h. These immune complexes had an elimination half-life of 5.5 h, mainly via the feces. 

Similarly, the entry of IgY into the brain across the blood–brain barrier was demonstrated, and it could neutralize the poison and act as an antivenom. The IgY antibodies specifically recognized the venom and as previously mentioned, did not activate the complement system, suggesting the possibility of safe re-administration until the resolution of intoxication symptoms in animals not developing ADE [60,61]. Since oral administration of IgY in humans has been shown to be effective and without side-effects, its parenteral administration needs further investigation to determine its safety, tolerability, and efficacy [62,63,64].

## 7. Conclusions

IgY technology allows the production of specific antibodies in large quantities according to international bioethical principles of reduction, refinement and, where possible, substitution of the use of experimental animals. These antibodies can be polyclonal or monoclonal against conserved mammalian proteins, with infrequent cross-reactions in diagnosis, since they are not the targets of rheumatoid factor. In addition, they do not induce complement activation, so they can be used for treatment and prophylaxis. These characteristics and their versatility make IgY antibodies powerful biotherapeutic molecules (Table 1).

## Figures and Tables

**Figure 1 antibodies-11-00062-f001:**
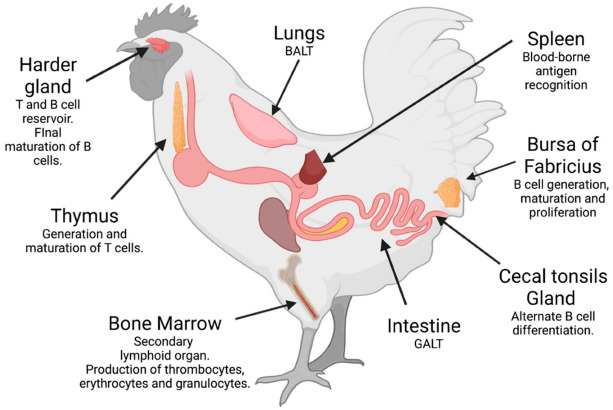
Organs of the immune system of birds and their functions, including the mucous-associated lymphoid tissues (MALTs), named the gut-associated (GALTs) and the bronchus-associated lymphoid tissues (BALTs). Created with BioRender.com (accessed on 8 August 2022).

**Figure 2 antibodies-11-00062-f002:**
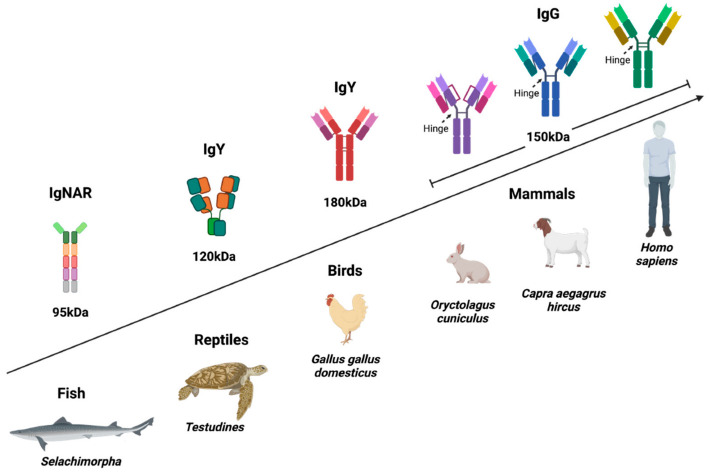
Structure of IgY compared to the IgG of different species. Avian IgY differs from mammalian IgG in that it has four constant domains and lacks a hinge region, which gives it high specificity. The Fc of IgY presents greater resistance to proteolytic degradation, does not bind to rheumatoid factor, and does not activate the human complement. Some reports suggest that IgY can increase the phagocytosis of specific pathogens, which could favor their elimination, but it is also possible that it promotes the infection of cells involved in the immune response, favoring their maintenance [13]. Therefore, more clinical studies are required to validate the safety and effectiveness of IgY in humans. Created with BioRender.com (accessed on 8 August 2022).

**Table 1 antibodies-11-00062-t001:** IgY technologies show advantages compared to antibodies from mammals.

Area of Advantage	IgY Advantage	Reference
Animal care	Replaces the bleeding of the animals by the collection of eggs, promoting animal welfare and reducing suffering associated with painful handling	[16,17,18,19]
Biotechnology, diagnosis, and human treatment	Low-cost production	[13,14,15,16,17,18,19,20,21,22,23,24,55,56]
	Antibodies against mammal conserved proteins	
	High-yield of production	
	High specificity	
	Can be used as polyclonal, monoclonal, Fab, or scFv	
	IgY Fc does not interact with Fc receptors	
	IgY Fc does not activate complement	
Human treatment	Toxic side-effects not yet reported; if used in pure form they can even be used in patients with allergy to eggs	[55,56]
	Passive immunization technology is applicable in newborns and adults, in immunodeficient patients and in pregnant women	

## Data Availability

No new data were created or analyzed in this study. Data sharing does not apply to this article.

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
