# Peer review of "IgY Antibodies as Biotherapeutics in Biomedicine"

_2073-4468, 2022, doi:10.3390/antib11040062_

Round 1

Reviewer 1 Report

In their review, León-Núñez and collegues go over the history and highlights of the possible utilization of IgY for biomedical research and applications. One negative aspects I have are the number of similar reviews in the last year – however, this one has the benefit of being short and to the point.  The use of the English langue could be improved, and some of the text feels more like a list than story telling. The more specific comments are listed below.

-Fig 1: GALT not fully spelled out. BALT not shown. What about general MALT

-3 IgG generally considered to be 150 kDa, although some subclasses such as human IgG3 is a little larger

-p2, “IgY has four constant domains, while IgG has only three. “ I think you need to clarify that you mean “immunoglobulin” domains. In essence this is partly incorrect as human IgG also has four domains as exons, three of which are immunoglobulin domains. Can you say something about exonal structures as structures of IgE and compare with human/mammalian IgG and IgE which in fact also have four constant exons immunoglobulin, of which 1 is the hinge of IgG. IgE and IgM lacks the hinge exon which instead have the forth immunoglobulin fold domain, much like IgY.

-lines 99-102: can you say something about background due to Fc-receptor binding? This is a problem for some mouse isotypes using human cells, but I assume this is much less of a problem for chicken IgY. Is there anything know about this?

-The same is highly relevant in the discussions regarding therapeutic use in lines 198, that should also resonance with the text on ADE.

-lines 141-44: rather awkward formulation. Reformulate in a more objective way will help. In fact what follows in this whole section is a rather a simple list (and sometimes repetitive), which is a missed opportunity as this contains interesting facts that can form a good story. Please reformulate without repetitions.

Author Response

In their review, León-Núñez and collegues go over the history and highlights of the possible utilization of IgY for biomedical research and applications. One negative aspects I have are the number of similar reviews in the last year – however, this one has the benefit of being short and to the point.  The use of the English langue could be improved, and some of the text feels more like a list than story telling. The more specific comments are listed below.

Response: Thank you for the positive comments. We agree that there are several recent reviews, but as the reviewer states, this is a concise text, with the aim to make it accessible to readers that might be interested in the significance of these antibodies but who prefer not to read long documents.

Regarding English spelling and grammar, we made a careful revisions throughout the text. Thank you.

-Fig 1: GALT not fully spelled out. BALT not shown. What about general MALT

Response: Please review lines 47 - 49 of the actual version; we added the MALT general concept (spelled out first) and specified GALTs; BALT was already present.   -3 IgG generally considered to be 150 kDa, although some subclasses such as human IgG3 is a little larger.

 Response: The reviewer is right. We changed the IgG size to 150KDa in figure 2 and in lines 77

-p2, “IgY has four constant domains, while IgG has only three. “ I think you need to clarify that you mean “immunoglobulin” domains. In essence this is partly incorrect as human IgG also has four domains as exons, three of which are immunoglobulin domains. Can you say something about exonal structures as structures of IgE and compare with human/mammalian IgG and IgE which in fact also have four constant exons immunoglobulin, of which 1 is the hinge of IgG. IgE and IgM lacks the hinge exon which instead have the forth immunoglobulin fold domain, much like IgY.

Response: We appreciate the comment of the reviewer, we have added the suggested information in lines 79 to 81.

-lines 99-102: can you say something about background due to Fc-receptor binding? This is a problem for some mouse isotypes using human cells, but I assume this is much less of a problem for chicken IgY. Is there anything know about this?

Response: In reference 13, there is mention that IgY enhances phagocytosis, but it was reported as non-dependent on FcR binding. This idea was added in the text in lines 83-84.

-The same is highly relevant in the discussions regarding therapeutic use in lines 198, that should also resonance with the text on ADE.

Response: We agree. We added the following sentence in lines 243 - 244.

-lines 141-44: rather awkward formulation. Reformulate in a more objective way will help. In fact what follows in this whole section is a rather a simple list (and sometimes repetitive), which is a missed opportunity as this contains interesting facts that can form a good story. Please reformulate without repetitions.

Response: we changed the way the paragraph is written. We hope it is better now.

Reviewer 2 Report

Authors show a brief summary on IgY antibody and IgY antibodies relatives. The subject is interesting, but at the moment this topic is well covered in several reviews. Very recent reviews (eg: Scientometric analysis and perspective of IgY technology study, Poult Sci 101, no. 4 229 (2022): 101713.) also cited in this review, describe in a much more complete way the importance of the use of IgY antibody. This review overlaps with other reviews reported in literature and therefore, I am against it.

Furthermore, the review does not touch upon important issues about the IgY production such as:

-Standard dogma in biopharmaceutical manufacturing is to use easy-to-remove, validatable reagents with a clear adherence to Good Manufacturing Principles. For example, the IgY used during manufacture of a biopharmaceutical must be themselves produced as pharmaceutical-grade IgY and need a clear strategy to remove relative impurities from the drug substance or drug product. While this is very briefly alluded to in the document, it needs more clarification and discussion in the paper. For example, questions that need to be addressed or discussed are:

1) would the presence of trace IgY-related impurities in a biopharmaceutical raise toxicological or patient safety concerns (are there any published data?), as it is difficult to guarantee 100% removal from the product.

2) strategies for removal of endotoxin or HCP from the product.

3) methods to show absence of IgY-related impurities from the product, and validation strategies.

- Also, a detailed description of advantages and disadvantages of the IgY technology should be better addressed, describing recent data.

Again, while the subject matter is interesting and the broad organization is acceptable, however the key content is not enough. I don't recommend this manuscript because for me to re-consider this paper significant additions of text and figures would need to be made.

Author Response

Authors show a brief summary on IgY antibody and IgY antibodies relatives. The subject is interesting, but at the moment this topic is well covered in several reviews. Very recent reviews (eg: Scientometric analysis and perspective of IgY technology study, Poult Sci 101, no. 4 229 (2022): 101713.) also cited in this review, describe in a much more complete way the importance of the use of IgY antibody. This review overlaps with other reviews reported in the literature and therefore, I am against it.

Response:

We appreciate the reviewer's comments.

We think that the paper mentioned by the reviewer (https://doi.org/10.1016/j.psj.2022.101713) is a very interesting paper extensively focusing on the scientometric analysis of the structure and the evolution trend of IgY technology over time, assessing the research effect, exploring the impact of academic journals and research institutions in this area of knowledge, and including analysis of techniques for citation inter-relationships. The authors conclude that IgY technology has made significant progress in the last 2 decades and has proven that it can be applied for diagnostic, prophylactic, or treatment purposes. In contrast, as the reviewer mentioned, our manuscript intends to be a concise summary of recently reported advances in IgY technology, especially in the diagnosis and possible treatment of human and animal diseases.

Furthermore, the review does not touch upon important issues about the IgY production such as:

-Standard dogma in biopharmaceutical manufacturing is to use easy-to-remove, validatable reagents with a clear adherence to Good Manufacturing Principles. For example, the IgY used during manufacture of a biopharmaceutical must be themselves produced as pharmaceutical-grade IgY and need a clear strategy to remove relative impurities from the drug substance or drug product. While this is very briefly alluded to in the document, it needs more clarification and discussion in the paper. For example, questions that need to be addressed or discussed are:

1) would the presence of trace IgY-related impurities in a biopharmaceutical raise toxicological or patient safety concerns (are there any published data?), as it is difficult to guarantee 100% removal from the product.

2) strategies for removal of endotoxin or HCP from the product.

3) methods to show absence of IgY-related impurities from the product, and validation strategies.

- Also, a detailed description of advantages and disadvantages of the IgY technology should be better addressed, describing recent data.

Again, while the subject matter is interesting and the broad organization is acceptable, however the key content is not enough. I don't recommend this manuscript because for me to re-consider this paper significant additions of text and figures would need to be made.

Response: We have changed the manuscript according the reviewer suggestions. Included in the new section 5.

Reviewer 3 Report

The authors have made a successful attempt at summarizing the current biotherapeutic landscape of IgY antibodies and have time and again discussed potential future possibilities and few caveats at using this format for human administration.

- Having said that, a similar review on IgY (although not under such a broad title) has been published in 2020 in Vaccines journal (MDPI) under the heading,
"Can Immunization of Hens Provide Oral-Based Therapeutics against COVID-19?"
https://doi.org/10.3390/vaccines8030486

Can the authors cite this review as well?

- Also, I would like to draw the author's attention to lines 151-154 (with ref.#45) which are exactly same as the lines 146-148? Can you explain the purpose of this exact repetition?

- Also, in the line 201, do you mean peritoneal administration?

- Can the authors comment on the following;
a. Developability profiles of the IgY antibodies in general
b. Inter-animal variation on IgY production in poultry. MHC haplotypes responsible for higher or lower antibody titers
c. Purification methods of these molecules (as IgYs are dispered in yolk-lipid emulsion and IgYs cannot be purified by protein A/G)
d. Emergence of transgenic chicken technology (like OmniChicken from Ligand Pharmaceuticals) and its significance in this context

Author Response

The authors have made a successful attempt at summarizing the current biotherapeutic landscape of IgY antibodies and have time and again discussed potential future possibilities and few caveats at using this format for human administration.

- Having said that, a similar review on IgY (although not under such a broad title) has been published in 2020 in Vaccines journal (MDPI) under the heading,
"Can Immunization of Hens Provide Oral-Based Therapeutics against COVID-19?"
https://doi.org/10.3390/vaccines8030486

Can the authors cite this review as well?

Response: Thank you for the suggestion, we have included it.

- Also, I would like to draw the author's attention to lines 151-154 (with ref.#45) which are exactly same as the lines 146-148? Can you explain the purpose of this exact repetition?

Response: We apologize for our mistake in preparing the manuscript, now we have changed this paragraph.

- Also, in the line 201, do you mean peritoneal administration?

Response: no, we meant parenteral; we thank the reviewer for detecting this big error, already corrected.

Can the authors comment on the following;
a. Developability profiles of the IgY antibodies in general 
b. Inter-animal variation on IgY production in poultry. MHC haplotypes responsible for higher or lower antibody titers
c. Purification methods of these molecules (as IgYs are dispered in yolk-lipid emulsion and IgYs cannot be purified by protein A/G)
d. Emergence of transgenic chicken technology (like OmniChicken from Ligand Pharmaceuticals) and its significance in this context

Response: Thank you for your suggestions, we have included the info in the manuscript, particullarly in the new section 5.

Round 2

Reviewer 2 Report

The author addressed in reviewer's comment accordingly. The reviewer does not have additonal comments.